# Prediction Model for Hypertension and Diabetes Mellitus Using Korean Public Health Examination Data (2002–2017)

**DOI:** 10.3390/diagnostics12081967

**Published:** 2022-08-14

**Authors:** Yong Whi Jeong, Yeojin Jung, Hoyeon Jeong, Ji Hye Huh, Ki-Chul Sung, Jeong-Hun Shin, Hyeon Chang Kim, Jang Young Kim, Dae Ryong Kang

**Affiliations:** 1Department of Biostatistics, Wonju College of Medicine, Yonsei University, Wonju 26426, Korea; 2Department of Medicine, Wonju College of Medicine, Yonsei University, Wonju 26426, Korea; 3Division of Endocrinology and Metabolism, Department of Internal Medicine, Hallym University Sacred Heart Hospital, Anyang 14068, Korea; 4Division of Cardiology, Department of Internal Medicine, Kangbuk Samsung Hospital, Sungkyunkwan University School of Medicine, 29 Saemunan-ro, Jongno-gu, Seoul 03181, Korea; 5Division of Cardiology, Department of Internal Medicine, Hanyang University College of Medicine, Seoul 04763, Korea; 6Department of Preventive Medicine, Yonsei University College of Medicine, 50-1, Yonsei-ro, Seodaemun-gu, Seoul 03722, Korea; 7Division of Cardiology, Wonju College of Medicine, Yonsei University, Wonju 26426, Korea; 8Department of Precision Medicine, Wonju College of Medicine, Yonsei University, Wonju 26426, Korea

**Keywords:** health check-up, hypertension, diabetes mellitus, logistic regression, random forest, XGBoost

## Abstract

Hypertension and diabetes mellitus are major chronic diseases that are important factors in the management of cardiovascular disease. In order to prevent the occurrence of chronic diseases, proper health management through periodic health check-ups is necessary. The purpose of this study is to determine the incidence of hypertension and diabetes mellitus according to the health check-up, and to develop a predictive model for hypertension and diabetes according to the health check-up. We used the National Health Insurance Corporation database of Korea and checked whether hypertension or diabetes occurred from that date according to the number of health check-ups over the past 10 years. Compared to those who underwent five health check-ups, those who participated in the first screening had hypertension (OR = 2.18, 95% CI = 2.14–2.22), diabetes mellitus (OR = 1.33, 95% CI = 1.30–1.35) and both diseases (OR = 2.46, 95% CI = 2.39–2.53); individuals who underwent 10 screenings had hypertension (OR = 0.86, 95% CI = 0.83–0.88), diabetes mellitus (OR = 0.83, 95% CI = 0.81–0.85) and both diseases (OR = 0.83, 95% CI = 0.79–0.87). Individuals who attended fewer than five screenings compared with individuals who attended five or more screenings had hypertension (OR = 1.61, 95% CI = 1.59–1.62; AUC = 0.66), diabetes mellitus (OR = 1.21, 95% CI = 1.20–1.22; AUC = 0.59) and both diseases (OR = 1.75, 95% CI = 1.72–1.78, AUC = 0.63). The machine learning-based prediction model using XGBoost showed higher performance in all datasets than the conventional logistic regression model in predicting hypertension (accuracy, 0.828 vs. 0.628; F1-score, 0.800 vs. 0.633; AUC, 828 vs. 0.630), diabetes mellitus (accuracy, 0.707 vs. 0.575; F1-score, 0.663 vs. 0.576; AUC, 0.710 vs. 0.575) and both diseases (accuracy, 0.950 vs. 0.612; F1-score, 0.950 vs. 0.614; AUC, 0.952 vs. 0.612). It was found that health check-up had a great influence on the occurrence of hypertension and diabetes, and screening frequency was more important than other factors in the variable importances.

## 1. Introduction

High blood pressure and diabetes are major modifiable risk factors of cardiovascular disease (CVD), the leading cause of death globally [1]. In rapidly aging societies, greater burden from CVD is expected, making the control and management of these modifiable risk factors especially important [2]. Research indicates that while the prevalence of high blood pressure has decreased over the past few decades, especially among high-income Western and Asia Pacific countries, global mean blood pressure levels have remained virtually unchanged [3]. Moreover, the prevalence of diabetes has been on the rise: although increases have been relatively gradual in the European region, several countries have recorded drastic increases in prevalence, especially in the Oceania region [4]. In Korea, the age-standardized prevalence of hypertension among Korean adults over the years 2001 to 2011 did not rise over 24% and was lowest in 2007 at 19.7%. Meanwhile, trends of increasing prevalence for diabetes and impaired fasting glucose were recorded among Korean adults aged 30 years or older, with a diabetes prevalence of 9.9% in 2007–2009. The prevalence of hypertension and diabetes both increased with age [5].

Studies in the literature suggest a need for more stringent management for uncontrolled risk factors related with CVD [5,6,7]. Currently, the US Preventive Services Task Force recommends yearly screening for hypertension among high-risk individuals and adults 40 years and older and less frequent screening for younger adults with low risk [8]. Overweight or obese adults of ages 35 to 70 years are also recommended to receive regular screening for diabetes [9]. In Korea, all adults older than 19 years are eligible for general health screening every 2 years via the National Health Screening Program, which from 2009 began to focus on reducing cardiovascular and cerebrovascular diseases via health risk assessment and lifestyle modification [10,11]. The program consists of seven categories, including history taking and measurements of height, weight, body mass index (BMI), blood pressure, hemoglobin, glucose, lipids, liver function, and renal function. In 2009, 63.3% of about 16 million people who were eligible for the general health screening participated [12]. Participation rates have climbed steadily since, reaching 72.3% in 2014 and 74.1% in 2019.

Several studies have examined the relationship between health check-up participation and cardiovascular risk factors [13,14,15], finding that receiving health test interventions are associated with lowered risk scores and better life expectancy [15], lower risks of all-cause mortality and cardiovascular events [14], and lower prevalence of several diseases, including metabolic syndrome [13], and that screening interval can affect health outcomes [16,17]. Meanwhile, however, some studies have indicated that screening participation has no effect on changes in disease prevalence in comparison to only receiving usual care [18]. Nevertheless, there is not yet enough data on whether the frequency of participation in optional health examinations can affect cardiometabolic risk factors, and few longitudinal studies have explored the effects of health examinations over an expansive time frame.

Therefore, in this study, we aimed to determine the relationship between screening frequency and new incidences of hypertension and diabetes utilizing national screening data collected over a 10-year period. We also set out to build a model based on these findings that can accurately predict the incidence of hypertension and diabetes.

## 2. Methods

### 2.1. Study Design

The data of this study are longitudinal and were captured over a baseline period from 2002 to 2011. Depending on the number of health check-ups over the past 10 years, long-term follow-up was performed to determine whether hypertension or diabetes occurred from the date of the last checkup to 2017. To accurately determine whether hypertension or diabetes occurred, all subjects with hypertension or diabetes before the last examination were removed. In consideration of the fact that national health check-ups can be received once every 2 years from the age of 19 years or older and the 10-year baseline period, the age of the subjects was set at 30 years or older, and subjects who underwent more than five check-ups were grouped separately from those with fewer than five check-ups to compare the incidences of hypertension and diabetes (Figure 1).

### 2.2. Dataset

The dataset used in this study was extracted from the National Health Insurance Service as a customized research database from 2002 to 2017, and the data reflected subjects who received health check-ups from 2002 to 2011 (*n* = 2,229,319). Among them, subjects with hypertensive disease codes (I10–I15) or who were taking antihypertensive drugs (Appendix A) and diabetes disease codes (E11–14) or who were taking diabetes medications (Appendix A) before the last screening were excluded (*n* = 1,502,179). Considering that national health check-ups are provided every 2 years for all people over the age of 19, subjects under the age of 30 were removed (*n* = 59,048), along with subjects with missing values in the database (*n* = 58,517) and subjects who died before 2011 (*n* = 162,431). In consideration of the fact that national health check-ups are provided every 2 years, the subjects were divided into those who received more than five health check-ups (*n* = 330,517) and those who received fewer than five health check-ups (*n* = 983,976). After propensity score matching (PSM), the final dataset consisted of 330,517 subjects who received more than five health check-ups or fewer than five health check-ups (Figure 2).

### 2.3. Measurements and Definition

The independent variables used to determine the characteristics related to the occurrence of hypertension and diabetes mellitus according to the number of examinations were as follows: age, sex, income, BMI, diastolic blood pressure, systolic blood pressure, fasting blood sugar, total cholesterol, alcohol consumption, smoking, and physical activity. Income level was calculated based on the insurance owner’s income level to claim health insurance premiums and was classified into quartiles. Body mass index (BMI) was estimated as body weight (kg) divided by height squared (m^2^). Blood samples were taken during the health examination after an overnight fast of at least 8 h and total cholesterol was enzymatically assessed. Alcohol consumption was divided into four categories grouped according to the frequency for weekly consumption of alcohol. Smoking was divided into 3 categories according to current smoking status. Physical activity was assessed in accordance with Metabolic Equivalent of Task definitions [19].

### 2.4. Study Outcomes

The outcomes of this study were hypertension, diabetes mellitus, and both diseases (hypertension and diabetes mellitus). Hypertension was defined as a blood pressure ≥ 140/90 mmHg or at least one claim per year for antihypertensive medication prescription under ICD-10 codes I10–I15. Diabetes mellitus was defined as type 2 diabetes with an FPG level ≥ 126 mg/dL or at least one claim per year for the prescription of hypoglycemic drugs under ICD-10 codes E11–14 [20]. Having both diseases was defined as the occurrence of both hypertension and diabetes mellitus within the follow-up period. The observation period was set from 1 January 2002 to 31 December 2011, and the follow-up period was set from 1 January 2012 to 31 December 2017. However, within the observation period, the final examination date may vary for each subject. This period can range from a minimum of 6 years to a maximum of 16 years, as the study design is followed-up from the date of final screening.

### 2.5. Statistical Analysis

Categorical variables are described as numbers and percentages. Continuous variables that followed normal distribution are summarized as means and standard deviations. To compare the baseline characteristics of the participants, an independent t-test was used for parametric continuous variables, and the Wilcoxon rank-sum test was used for non-parametric variables. The chi-square test was used for categorical variables. To reduce the effect of selection bias and potential confounders, we adjusted for significant differences in the baseline characteristics of subjects using 1:1 PSM with caliper set to 0.25 [21]. As variables for PSM, age, sex, systolic blood pressure, diastolic blood pressure, and fasting blood glucose, which are variables directly related to hypertension and diabetes mellitus, were used. A value corresponding to 1⁄4 of the standard error of the estimated propensity score was designated as a range and used for pairing. Only when the difference in propensity score between paired subjects falls within this range, pairing is included in the analysis. All excluded subjects were excluded from the analysis. All PSM procedures were performed, and we compared baseline covariates between the groups. Continuous variables were compared using paired t tests or Wilcoxon signed-rank tests, as appropriate, and categorical variables were compared using McNemar’s test. Univariate logistic regression and multiple logistic regression analyses were used to evaluate the relationship between screening frequency and the main outcomes and the relationship between explanatory variables and the main outcomes. All *p* values less than 0.05 were considered statistically significant, and standardized differences of covariates used in PSM analysis were considered significant when less than 0.1.

Logistic regression, random forest, and XGBoost were used to develop the predictive model. Logistic regression is a probability model that can be explained by a model using relationships between independent and dependent variables. Random forest creates a large number of decision trees by randomly sampling training data and then uses the results of the decision trees to derive a final result [22]. XGBoost was developed using the negative slope of loss function as the residual value of the current fitting to achieve an accurate classification effect. XGBoost reduces overfitting by performing a quadratic Taylor extension of the loss function and adding a regular term outside the loss function to balance the reduction of the loss function with the complexity of the model [23].

We divided the full dataset into a training dataset (70%) and a test dataset (30%) using random sampling. Since the range for each independent variable differed, all of the variables were normalized with the minimum and maximum values of each variable in the training dataset. We used a grid search to determine the optimal hyper-parameters and performed 5-fold cross validation to prevent overfitting. Since all datasets comprise imbalanced data, the proportion of unbalanced data was made the same using the synthetic minority oversampling technique [24]. In order to evaluate the performance of each prediction model, we used receiver operating characteristics curves, area under the curve (AUC), F1 score, and accuracy. The software used for the analyses were as follows: SAS 9.4 (SAS, Cary, NC, USA), Python 3.5.2, pandas 0.24.2, sklearn 0.20.3, numpy 1.16.2, matplotlib 3.0.3, and scipy 1.2.1.

## 3. Results

The baseline characteristics of the study subjects are presented in Table 1. We found that participation rates in the health screening program were lower among women than men. Among subjects who underwent screening less than five times, systolic blood pressure, fasting blood glucose, total cholesterol, rate of drinking more than half a week, and the proportion of subjects currently smoking were high; physical activity was very low; and the incidences of hypertension alone, diabetes alone, and both diseases were high.

Figure 3 presents the odds ratios and 95% confidence intervals for relationships between screening frequency and hypertension and diabetes. Individuals who participated in screening once were associated with a significantly increased odds for having hypertension (OR = 2.18, 95% CI = 2.14–2.22), diabetes mellitus (OR = 1.33, 95% CI = 1.30–1.35), and both diseases (OR = 2.46, 95% CI = 2.39–2.53) compared to the nationally recommended screening frequency of five. Individuals who underwent screening 10 times were associated with a significantly lower odds for having hypertension (OR = 0.86, 95% CI = 0.83–0.88), diabetes mellitus (OR = 0.83, 95% CI = 0.81–0.85), and both diseases (OR = 0.83, 95% CI = 0.79–0.87) compared to the nationally recommended screening frequency of five. We performed sensitivity analyses according to the presence of hypertension and diabetes mellitus and screening frequency (Appendix A). Appendix A displays age groups stratified by 10 years and sex, and when the adjusted odds ratio and odds ratio values were compared, the same trends noted above were observed without any difference.

The ORs and confidence intervals for hypertension (OR = 1.61, 95% CI = 1.59–1.62; AUC = 0.66), diabetes mellitus (OR = 1.21, 95% CI = 1.20–1.22; AUC = 0.59), and both diseases (OR = 1.75, 95% CI = 1.72–1.78; AUC = 0.63) among individuals who attended screening less than five times and individuals who attended screening five times or more as a reference are shown in Table 2.

Table 3 shows the results of the evaluation of each model. Random forest and XGBoost performed better than logistic regression models for all metrics in all datasets, and XGBoost performed the best. In particular, the model for predicting hypertension and diabetes showed the highest performance (accuracy, 0.950; $f_{1}$-score, 0.950; AUC, 0.952 [0.951–0.953]). Figure 4 comprises a comparison of receiver operating characteristics curves for the logistic regression model, random forest, and XGBoost for each dataset and supports the results shown in Table 3. The variables with the highest importance in most models included age, screening frequency, sex, smoking, and BMI. Of these, age and screening frequency had the greatest effect on model performance.

## 4. Discussion

Previous studies on the impact of participation in health screening programs have offered varying results. One randomized trial reported that health test participation was associated with reduced cardiovascular risk score [15], while another showed that population screening increased the detection of CVD risk factors, such as hypertension [25]. General screening has been shown to be associated with increased detection of chronic disease in several other studies [14,18,26,27,28,29,30]. A review article also reported that general health checks were associated with increased detection and treatment of chronic diseases, but had little impact on mortality or cardiovascular event reductions [31]. However, these results remain controversial as other studies have reported that health check participation is associated with fewer cardiovascular risk factors [13], risk factor value reductions [32], and mortality reductions [33]. Additional studies have been conducted to identify social elements that can affect participation, such as perceived susceptibility and health knowledge [34,35], socioeconomic status [36,37], medical history [38,39,40,41], health behavior factors, and sociodemographic factors [42,43]. Despite the above reasons, there have not been many studies on chronic disease prediction models according to health check-ups, so it is necessary to confirm new facts using large-scale data.

A hypothesis that the incidences of hypertension and diabetes would differ in two groups divided according to the number of health examinations could be established due to the lack of studies on the relationship between health check-ups and chronic diseases and the fact that a lot of information was available in the NHIS data. However, since an accurate verification method was needed to prove our hypothesis, we applied two research methods: confirmatory and exploratory. In order to conduct a confirmatory study, age, sex, systolic blood pressure, diastolic blood pressure, and fasting blood glucose, which may act as confounding variables, were examined to evaluate the hypothesis. In logistic regression analysis to compare the incidences of hypertension and diabetes according to the two groups, age and sex, as well as the number of health examinations, were adjusted, and sensitivity analysis was used to increase the reliability of the results. In order to proceed with the exploratory study, a classification model was developed using various variables, and the importance of the top five variables among the variables used in the model was extracted to confirm the relationships between the variables used in our studies.

There are many previous studies that predict diabetes or hypertension. People with a family history of diabetes are known to have a higher risk of diabetes than people without a family history of diabetes [44]; most of the family history variables are used in the diabetes predictive model [45,46]. However, family history variables were not considered in our study for the following reasons. Since the main purpose of this study is to explore whether or not health check-ups have a significant effect on the occurrence of chronic diseases, the predictive model requires a large number of subjects. Using family history of diabetes as a variable in the NHIS data has the disadvantage of a significant decrease in subjects (Appendix A). However, if it is simply for the above reasons, there may be a prejudice that it is to show that the results of our study are good. Therefore, logistic regression analysis was performed, and there was no difference between the results of adding a family history of diabetes and the results obtained in the study, AUC, and odds ratio (Appendix A). The results of logistic regression analysis confirmed that the probability of high blood pressure and diabetes decreased as the number of health examinations increased. Similar results have been observed in previous studies [14,15,18,26,27,28,29,30]. Upon extracting the importance of the variables used in the analyses and the variables used in the modeling, we confirmed that the number of health examinations and age were the most important variables in several models.

In terms of general health screening, the Korean government has not only continued expanding the eligible population and facilitating participation but has also implemented policies to ensure improvements in quality, equity, and result management in the national health screening program [47]. One of these policies is the third comprehensive National Health Screening Program plan [48], which was announced in 2021. In this, the government planned to promote self-health management by providing more in-depth information on individual health screening results, regional screening center locations, and reservation notifications through mobile applications. Other laws are also in place to encourage follow-up hospital visits after general check-ups, subsidizing medical expenses incurred by examinations for definite diagnosis in people who have taken national health examinations [49]. Such efforts of the Korean government and the medical community have led to a rise in health screening follow-up management rates from 6.0% in 2008 to 13.9% in 2018 [50].

The results of this study showed that an increased participation in health screening was significantly related to the lowered occurrence of hypertension and diabetes. This suggests that encouraging the general population to participate diligently in health screenings may be an important factor to preventing disease and improving public health. Previous studies on the impact of screening on future health-related behaviors support this claim [51,52], stating that receiving the screening results may be a teachable moment for the participants, which then can lead to a change in lifestyle factors. In Korea, 76.9% of the insured population and 41.9% of recipients of medical care assistance participated in the 2018 national health check-up [53,54]. While these participation rates cover most of the population, there remain people who do not receive adequate screening for whichever reason. Therefore, it is important that national health check-ups be publicized and made easily accessible to all persons eligible for participation, regardless of socio-economic status or location. Encouraging health check-ups using the result and model from our study is a simple step that will help prevent chronic disease and will also help achieve the health goals set by the government.

There are some limitations to this study that should be taken into consideration. First is the possibility of self-selection bias, as it is entirely up to the individual to choose whether they participated in the screening program. Since we do not know the reasons for non-participation, the results of this study should be approached with caution. Second is the possibility of response bias in the self-report part of the screening program. Further randomized controlled trials are needed to clarify the results of health screening participation. Finally, in order to know how much predictive power depends on the number of health check-ups, a predictive model was constructed using logistic regression and machine learning technique. Better performance results could be obtained with using deep learning methods, such as long short-term memory and recurrent neural networks which can use risk factors that affect outcomes over time [46].

## 5. Conclusions

It was confirmed that the health check-up had a great influence on the occurrence of hypertension and diabetes. Periodic health check-ups are necessary to manage the occurrence of chronic diseases.

## Figures and Tables

**Figure 1 diagnostics-12-01967-f001:**
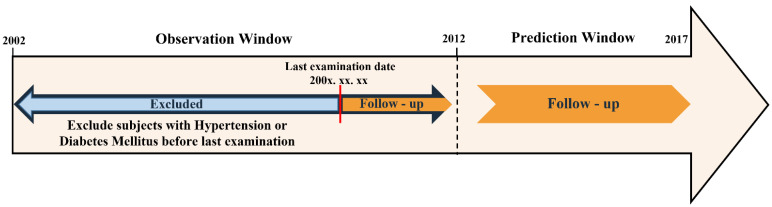
Study design.

**Figure 2 diagnostics-12-01967-f002:**
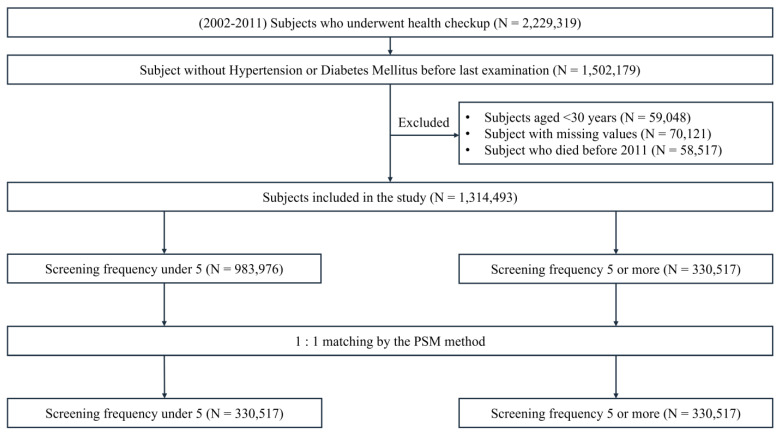
Flow chart of study design.

**Figure 3 diagnostics-12-01967-f003:**
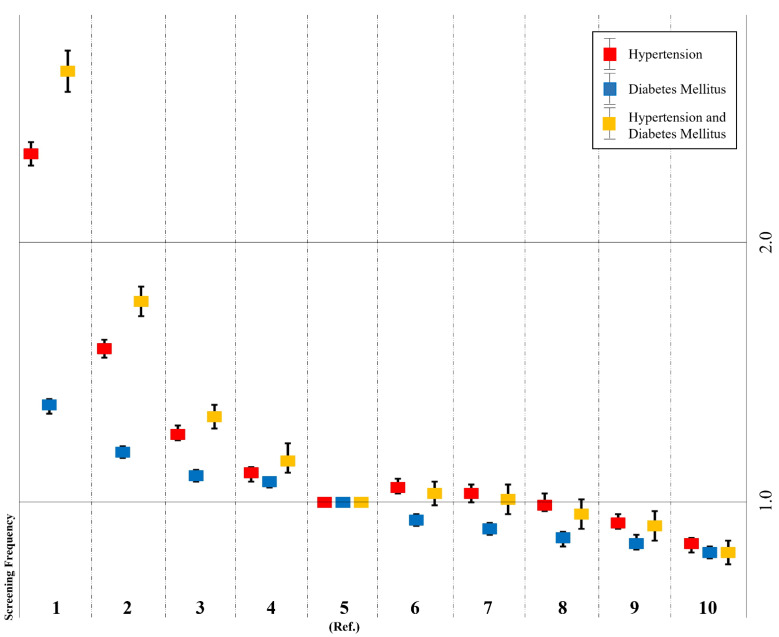
Odds ratios with 95% CIs for hypertension and diabetes mellitus according to screening frequency adjusted by age and sex.

**Figure 4 diagnostics-12-01967-f004:**
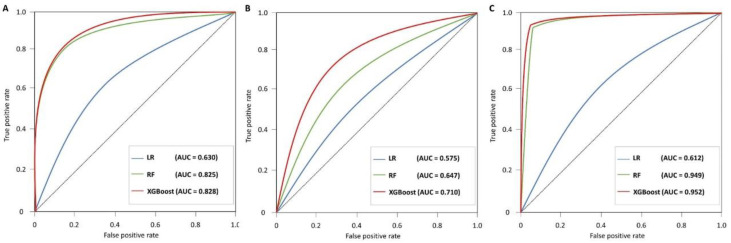
Comparison of receiver operating characteristics curves for (**A**) hypertension, (**B**) diabetes mellitus, and (**C**) hypertension and diabetes mellitus.

**Table 1 diagnostics-12-01967-t001:** Baseline Characteristics after Propensity Score Matching.

Variables	Total (*n* = 661,034)	Screening < 5 Times (*n* = 330,517)	Screening ≥ 5 Times (*n* = 330,517)	*p*-Value
Sex				0.4629
male	438,586 (66.35)	219,152 (66.31)	219,434 (66.39)	
female	222,448 (33.65)	111,365 (33.69)	111,083 (33.61)	
Age, years	53.79 (11.75)	53.83 (11.84)	53.77 (11.65)	0.3539
				<0.0001
30 s	80,435 (12.17)	39,861 (12.06)	40,574 (12.28)	
40 s	160,385 (24.26)	80,951 (24.49)	79,434 (24.03)	
50 s	209,695 (31.72)	103,871 (31.43)	105,824 (32.02)	
60 s	137,638 (20.82)	68,455 (20.71)	69,183 (20.93)	
70 s	72,881 (11.03)	37,379 (11.31)	35,502 (10.74)	
Income level				<0.0001
quartile 1	241,403 (36.52)	109,280 (33.06)	132,123 (39.97)	
quartile 2	173,063 (26.18)	87,821 (26.41)	85,782 (25.95)	
quartile 3	122,972 (18.60)	67,950 (20.56)	55,022 (16.65)	
quartile 4	123,596 (18.70)	66,006 (19.97)	57,590 (17.42)	
BMI, kg/m^2^				<0.0001
<18.5	11,321 (1.71)	6234 (1.89)	5087 (1.54)	
18.5–22.9	188,290 (28.48)	93,845 (28.39)	94,445 (28.57)	
23.0–24.9	173,636 (26.27)	84,532 (25.58)	89,104 (26.96)	
≥25.0	287,797 (43.54)	145,906 (44.14)	141,881 (42.93)	
Diastolic blood pressure, mmHg	80.68 (10.39)	80.66 (10.42)	80.71 (10.36)	0.0473
Systolic blood pressure, mmHg	129.18 (15.11)	129.16 (15.19)	129.20 (15.02)	0.2487
Fasting blood sugar, mg/dL	98.14 (21.48)	98.16 (22.79)	98.12 (20.07)	0.4133
Total cholesterol, mg/dL	200.45 (29.69)	200.79 (41.43)	200.12 (37.86)	<0.0001
Alcohol consumption, times/week				<0.0001
0	322,171 (48.74)	164,182 (49.67)	157,989 (47.80)	
1	118,182 (17.88)	52,691 (15.94)	65,491 (19.81)	
2,3	153,556 (23.23)	72,804 (22.03)	80,752 (24.43)	
4–7	67,125 (10.15)	40,840 (12.36)	26,285 (7.95)	
Smoking				<0.0001
never	350,333 (53.00)	173,998 (52.64)	176,335 (53.35)	
ex	130,701 (19.77)	57,251 (17.32)	73,450 (22.22)	
current	180,000 (27.23)	99,268 (30.03)	80,732 (24.43)	
Physical activity, METs-min/week	953.57 (1227.30)	774.92 (1174.12)	1160.56 (1255.62)	<0.0001
Outcomes				
Hypertension				<0.0001
no	490,256 (74.17)	232,065 (70.21)	258,191 (78.12)	
yes	170,778 (25.83)	98,452 (29.79)	72,326 (21.88)	
Diabetes mellitus				<0.0001
no	400,243 (60.55)	191,519 (57.95)	208,724 (63.15)	
yes	260,791 (39.45)	138,998 (42.05)	121,793 (36.85)	
Hypertension and diabetes mellitus				<0.0001
no	603,723 (91.33)	294,778 (89.19)	308,945 (93.47)	
yes	57,311 (8.67)	35,739 (10.81)	21,572 (6.53)	

Data are presented as a *n* (%) or mean (SD).

**Table 2 diagnostics-12-01967-t002:** Logistic regression of hypertension and diabetes mellitus according to screening frequency group.

	Hypertension	Diabetes Mellitus	Hypertension and Diabetes Mellitus
Variable	OR (95% CI)	OR (95% CI)	OR (95% CI)
Screening frequency			
≥5 times	Ref.	Ref.	Ref.
<5 times	1.61 (1.59–1.62)	1.21 (1.20–1.22)	1.75 (1.72–1.78)

Adjusted by age and sex; OR, Odds Ratio; CI, Confidence Interval; Ref., reference.

**Table 3 diagnostics-12-01967-t003:** Model evaluation results.

Outcomes	Classifier	Accuracy	$f_{1}$-Score	AUC (95% CI)	Variable Importance *
Hypertension	Logistic Regression	0.628	0.633	0.630 (0.627–0.632)	**Age** > Screening frequency > Sex > BMI > Smoking
	Random Forest	0.824	0.798	0.825 (0.823–0.826)	**Age** > Screening frequency > Sex > Smoking > BMI
	XGBoost	**0.828**	**0.800**	**0.828** **(0.826–0.830)**	**Screening frequency** > Sex > Age > BMI > Smoking
Diabetes Mellitus	Logistic Regression	0.575	0.576	0.575 (0.572–0.578)	**Age** > Screening frequency > FBS > BMI > Sex
	Random Forest	0.693	0.647	0.647 (0.645–0.650)	**Age** > Screening frequency > FBS > BMI > Sex
	XGBoost	**0.707**	**0.663**	**0.710** **(0.708–0.712)**	**Screening frequency** > Sex > Age > BMI > Smoking
Hypertension and Diabetes Mellitus	Logistic Regression	0.612	0.614	0.612 (0.610–0.614)	**Screening frequency** > Sex > Age > Smoking >BMI
	Random Forest	0.948	0.946	0.949 (0.948–0.949)	**Screening frequency** > Smoking > Age > Sex > BMI
	XGBoost	**0.950**	**0.950**	**0.952** **(0.951–0.953)**	**Screening frequency** > Sex > Smoking > BMI > Age

* The five variables with the highest importance are indicated. Boldface means the highest value in each metric column. AUC, area under curve; CI, confidence interval; BMI, body mass index; FBS, fasting blood sugar.

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
