# Peer review of "Prediction Model for Hypertension and Diabetes Mellitus Using Korean Public Health Examination Data (2002–2017)"

_diagnostics, 2022, doi:10.3390/diagnostics12081967_

Round 1

Reviewer 1 Report

The manuscript presents data for the healthcare issue surrounding frequency of check-ups and incident hypertension and/or diabetes.   The manuscript is generally well written and logically presented, including the analytical steps.  The concerns are as follows:  

·      The title doesn’t fit the aim of the manuscript.  — suggest revising 

·      Authors state that the check-ups are designed for assessment and lifestyle modification, but this doesn’t provide sufficient context for the reader.  Is there a progressive healthcare strategy in Korea that outlines a visit dependent health action plan based on risk assessment? —suggest revising

·      The use of each of the three analytical strategies is well described in the methods and presented in the results, but their discussion is vague. Only the logistic regression mentioned by name.  As a big picture, Figure 3 provides the main data for the question of whether frequency of visits is important to risk. It also clearly shows that the goal recommendation of 5 visits in this Korea health check-up context is sound.  While it is clear from the ROC curves that more sophisticated modeling fine tunes what is presented with the logistic regression, there is no mention as to how this predictive modeling would be useful moving forward.  

Minor:

Flow diagram (Figure 2)  add dates to make consistent with Figure 1

Author Response

1. The title doesn’t fit the aim of the manuscript

Answer) 

  • Thanks for the advice. We Changed the title to “Prediction model for hypertension and diabetes mellitus using Korean Public Health Examination data(2002-2017)”.

2. Authors state that the check-ups are designed for assessment and lifestyle modification, but this doesn’t provide sufficient context for the reader.  Is there a progressive healthcare strategy in Korea that outlines a visit dependent health action plan based on risk assessment? —suggest revising

Answer) 

  • Thanks for pointing it out. Added what you said on lines 290-302 in the Discussion part(Please see the attachment).

3. The use of each of the three analytical strategies is well described in the methods and presented in the results, but their discussion is vague. Only the logistic regression mentioned by name.  As a big picture, Figure 3 provides the main data for the question of whether frequency of visits is important to risk. It also clearly shows that the goal recommendation of 5 visits in this Korea health check-up context is sound.  While it is clear from the ROC curves that more sophisticated modeling fine tunes what is presented with the logistic regression, there is no mention as to how this predictive modeling would be useful moving forward.  

Answer)

  • Thanks for telling me. Lines 303-317 were written in relation to that content(Please see the attachment).

4. Flow diagram (Figure 2) add dates to make consistent with Figure 1

Answer) 

  • Thanks for the advice. The observation period is from 2002.01.01 to 2011.12.31, and the follow-up period is from 2012.01.01 to 2017.12.31. The time of the last examination is different for each subject to be observed. Therefore, the observation period may vary slightly from subject to subject. To express it, it was expressed like figure1, and figure2 was explained in a large frame, and I inform you that the two contents are consistent.

Reviewer 2 Report

Some comments for the authors’ consideration:

·       The Abstract should be improved. Currently it doesn’t read very clearly what the reference group is with respect to which the odds ratios are derived. Consider changing the format of the Abstract to include the sections: Background, Motivation, Methods, Results, Conclusion.

·       Line 152 – PSM (Propensity Score Matching) needs to be described in more detail.

·       Line 158 – “main outcomes” are not clearly defined. Is it occurrence of hypertension, diabetes, and co-occurrence of hypertension and diabetes? If so, over which observation period? Consider designating a separate subsection (e.g., before sec.2.3) defining the estimands of interest with due level of detail and rigor.

·       Lines 162-171 – justification for the choice of the parameters for data analysis (percent allocation to training/validation sets, 5-fold cross-validation, etc.) should be given.

·       Table 1 – “Age, years” – two different p-values are given in the last column. This needs clarification.

·       Table 1 – “Physical activity, times/week” – the units don’t seem to be correct – please check/confirm.

·       Figure 3 – the title “Logistic regression for hypertension and diabetes mellitus according to screening frequency adjusted by age and sex” is confusing. Displayed are the odds ratios with 95% CI’s, so the title should mention this explicitly.

·       Figure 3 – what may be helpful is to provide more insight/characterization of the subgroups of subjects who had 1,2,3,…,10 screenings. Was there any particular baseline characteristic that tended to vary across these subgroups?

·     Consider adding some comments on the overall generalizability of the results – to which population do these findings apply?

Author Response

1. The Abstract should be improved. Currently it doesn’t read very clearly what the reference group is with respect to which the odds ratios are derived. Consider changing the format of the Abstract to include the sections: Background, Motivation, Methods, Results, Conclusion.

Answer) 

  • Thanks for pointing it out. A reference has been added to the abstract, and it has been modified according to the section you advised(Please see the attachment).

2. Line 152 – PSM (Propensity Score Matching) needs to be described in more detail.

Answer)

  • Thanks for telling me. Added more information about the psm method and matching variables on lines 163-174(Please see the attachment).

3. Line 158 – “main outcomes” are not clearly defined. Is it occurrence of hypertension, diabetes, and co-occurrence of hypertension and diabetes? If so, over which observation period? Consider designating a separate subsection (e.g., before sec.2.3) defining the estimands of interest with due level of detail and rigor.

Answer)

  • Thanks for the advice. As you pointed out, the contents of the study outcome have been added to Section 2.4. Section 2.4 includes the definition of outcome (hypertension, diabetes mellitus, both disease (hypertension and diabetes mellitus) and the observation period(Please see the attachment).

4. Lines 162-171 – justification for the choice of the parameters for data analysis (percent allocation to training/validation sets, 5-fold cross-validation, etc.) should be given.

Answer)

  • Let me tell you for the choice of the parameters for data analysis
  • At first, the data was split into an 8:2 ratio since the ratio of outcome was too high.
  • However, because the analysis was conducted based on the data of the entire nation, the characteristics of the subjects were too diverse. So the performance of the model was not good.
  • Also, the high ratio of the training set caused the model to overfit the data.
  • To address this issue, the following solutions were considered.
  • The model was first adjusted by lowering the training dataset ratio to 7:3, while using 5-fold cross validation to train the model with a higher variety of data to create a more generalized model.
  • Variables in the dataset were also normalized to account for the fact that each variable contained different characteristics.
  • Despite this process, the model performance AUC could not score over 0.6.
  • The last method was to use SMOTE to oversample/undersample the data to roughly match the outcome ratio. Of the two methods, oversampling proved superior.
  • Since SMOTE oversampling was used to build the model, the model shows strength in correctly identifying minor classes, but shows weaknesses predicting otherwise.
  • Nevertheless, the model scored higher in accuracy, F1 score, and AUC on the test dataset without oversampling, thus demonstrating the potential of our model to be a generalized model.

5. Table 1 – “Age, years” – two different p-values are given in the last column. This needs clarification.

Answer)

  • Thanks for telling me.
  • To answer the point you pointed out,
  • first of all, we used the continuous value of age during psmatching.
  • The reason for matching with a continuous variable is that matching may be possible in 1-year-old units.
  • If the difference between the two groups is used as a continuous variable, it can be said that there is no difference.
  • However, looking at the p-value by age group, it can be seen that there is a difference between the two groups.
  • Supplementary explanations are provided as to the reasons for these results.
  • First, since the above-mentioned matching was made in units of one year, there may be differences when the number of n is divided by group (10 years old). Also, since the number of n is very large, it can be considered a population and the p-value can be meaningless.
  • Second, the standardized differences of age used in PSM analysis was 0.008.
  • So you can see that there is a difference in the p-values, but I don't think it's a problematic factor.

6. Table 1 – “Physical activity, times/week” – the units don’t seem to be correct – please check/confirm.

Answer)

  • Thanks for pointing it out. I have changed the units correctly(Please see the attachment).

7. Figure 3 – the title “Logistic regression for hypertension and diabetes mellitus according to screening frequency adjusted by age and sex” is confusing. Displayed are the odds ratios with 95% CI’s, so the title should mention this explicitly.

Answer)

  • I agree with your opinion.
  • I edited with “Odds ratios with 95% CI’s for hypertension and diabetes mellitus according to screening frequency adjusted by age and sex”(Please see the attachment).

8. Figure 3 – what may be helpful is to provide more insight/characterization of the subgroups of subjects who had 1,2,3,…,10 screenings. Was there any particular baseline characteristic that tended to vary across these subgroups?

Answer)

  • We applied data to the National Health Insurance Corporation of Korea and used the data.
  • After the period of use is over, it is not possible to select subject who had 1,2,3,10 screenings and check the results.
  • However, there are resources that can help you get the results you want.
  • The content is shown in Figure s1.
  • After obtaining the results in Figure 3, sensitivity analyzes were performed to confirm that our results were correct.
  • To produce the results shown in Figure 3, age and sex were used as adjust variables in addition to the number of examinations.
  • There was no tendency to vary when adjusted for age and sex or analyzed as univariate.
  • So, as the results show, I think that the number of health checkups had a great influence on the results of figure 3(Please see the attachment).

9. Consider adding some comments on the overall generalizability of the results – to which population do these findings apply?

Answer)

  • Thanks for your advice. We have added what you said on lines 290-317 in the discussion part(Please see the attachment).

Round 2

Reviewer 1 Report

The manuscript has been improved with the revision.  No further concerns.

Reviewer 2 Report

The authors have addressed my comments.